# Parameter vs. Test-Time Scaling in LLMs: FLOPs-Aware, Cross-Domain, Domain-Dependent, Pareto-Optimal Compute Allocation

**Liner Research Agent**                Yungi Kim[1]        Sunghee Song[1]        Bumjun Jung[1*]
Liner Corp.
140, Yanghwa-ro, Mapo-gu,
Seoul, Republic of Korea 04050

[1]Liner Corp. (https://liner.com)
contact@linercorp.com

## Abstract

We study how to allocate compute between model size and test-time scaling (inference-time reasoning) to achieve cost-effective accuracy in large language models. We introduce a controllable-reasoning experimental design that directly compares parameter scaling and test-time scaling on mathematical reasoning (GSM8K) and knowledge retrieval (PopQA), using rigorous FLOPs and cost accounting and Gemini's thinking_budget to disentangle internal from external Chain-of-Thought (CoT) reasoning. Results show strong domain dependence. On GSM8K, internal reasoning alone reaches 95.36% accuracy at $3.8 \times 10^{-5}$ per sample, while CoT compensates for disabled internal reasoning to 95.60% at $9.4 \times 10^{-4}$, indicating near-perfect substitutability between internal and external mechanisms. On PopQA, external CoT often reduces both accuracy and cost-efficiency, with optimal settings consistently favoring direct generation over extended reasoning chains. We contribute: (1) the redundancy principle quantifying overlap between internal and external reasoning; (2) FLOPs-aware, domain-specific cost–accuracy Pareto frontiers that reveal distinct optimization strategies; and (3) actionable deployment policies that align test-time scaling with task characteristics and model architectures, providing evidence-based guidance for economical, high-performance LLM deployment.

## 1   Introduction

The rapid advance of large language models (LLMs) poses a core strategic question: should compute be invested in scaling model parameters or in test-time scaling via enhanced reasoning at inference? The answer has material economic consequences, with deployment costs differing by orders of magnitude. While parameter scaling follows established power laws [12, 11]—performance improving with model size, data, and training compute—techniques such as Chain-of-Thought (CoT) [25] prompting show that even smaller models can gain markedly through test-time reasoning [8]. For high-throughput applications, API pricing differentials (for example, around $30 versus $0.2 per 10 million tokens across model tiers [3]) make the choice between parameter and test-time scaling central to practical deployment.

Two paradigms dominate the literature yet remain insufficiently compared under matched compute: parameter scaling and test-time scaling. Early work showed predictable scaling on cross-entropy loss and in-context learning capabilities as parameters grow [13, 2], exemplified by GPT-3-sized models performing well without task-specific fine-tuning. In parallel, CoT prompting—providing

---

[*]Corresponding author

exemplars with intermediate reasoning steps—boosted performance on complex arithmetic [24, 14], commonsense [24, 31], and symbolic tasks [24], with pronounced gains appearing in large models (roughly ≥100B parameters) [24, 7]. However, prior studies rarely offer FLOPs-matched, cross-domain comparisons of these strategies or account for potential redundancy between internal latent reasoning and external prompting.

This work fills these gaps with a FLOPs-aware, cross-domain investigation of the parameter-versus-test-time scaling trade-off. Our central hypothesis is that optimal compute allocation depends on task structure and uncertainty. We posit that mathematical reasoning, which benefits from multi-step decomposition [7, 20], yields higher marginal returns to test-time scaling (e.g., CoT, majority voting) than to parameter scaling alone. In contrast, knowledge retrieval, which emphasizes access to parametric memory [17], should favor parameter scaling with lean prompting over elaborate reasoning [18, 19]. We further study a key but underexplored issue: redundancy between internal reasoning (latent computation) and external CoT, whereby strong models may already perform sufficient internal reasoning, rendering additional external CoT costly and largely unnecessary.

Our contributions are threefold. First, we provide a FLOPs-aware, cross-domain comparison of parameter versus test-time scaling and introduce a methodology to disentangle internal from external reasoning. Using Gemini's thinking_budget as a controllable toggle, we quantify redundancy and reveal domain-dependent returns to test-time compute: mathematical reasoning gains strongly from test-time scaling [30], whereas knowledge QA does not [26]. Second, we present a reproducible framework that standardizes prompts, metrics, and token-cost accounting across models and reasoning strategies, enabling precise cost–accuracy Pareto frontier analysis on GSM8K [5] and PopQA [16]. This addresses methodological inconsistencies that have impeded prior comparisons. Third, we derive deployment policies from the empirical frontiers. On mathematical tasks, CoT is effective for smaller models or when internal reasoning is disabled, fully compensating for reduced parametric capacity [21]; stacking CoT atop strong internal reasoning is economically inefficient. For knowledge retrieval, parameter scaling with lean prompting dominates, indicating that factual access benefits more from parametric memory than from extended inference computation [22].

Our scope spans representative model families—GPT-4.1 variants and the Gemini 2.5 series—allowing analysis across architectural paradigms and scales. Two constraints bound our conclusions: reliance on closed-source APIs for some evaluations and a focus on two domains (mathematical reasoning and knowledge retrieval). While these choices reflect common applications, they may not cover all task categories. Nevertheless, our study offers a systematic, empirically grounded framework for navigating the parameter-versus-test-time scaling dilemma, with immediate implications for cost-effective deployment where computational efficiency directly affects economics and user experience.

## 2 Related Work

Two principal paradigms shape the development of large language models: scaling parameters and augmenting test-time compute. Foundational studies show that performance, particularly cross-entropy loss, follows power-law relationships with model size, dataset size, and training compute [12]. This dynamic was exemplified by GPT-3-scale models, where billions of parameters enabled strong few-shot learning via in-context examples without task-specific fine-tuning [2]. A key observation is that larger models demonstrate markedly improved in-context learning, with few-shot gains accelerating relative to zero-shot as parameters increase [2, 28].

In parallel, Chain-of-Thought (CoT) prompting emerged as a powerful form of test-time scaling: providing exemplars with intermediate reasoning steps can significantly improve complex arithmetic, commonsense, and symbolic reasoning [25, 23]. Crucially, such benefits were most pronounced in larger models (approximately ≥100B parameters), while smaller models often produced incoherent or brittle reasoning traces [15]. The combination of extreme parameter scale and CoT—as in very large models—revealed discontinuous jumps in multi-step reasoning capability, linking scale to potential for effective test-time reasoning.

Subsequent work has sought greater efficiency through adaptive computation [6, 10] and inference-cost-aware strategies [27]. Examples include mechanisms that allow models to allocate more internal compute on harder problems (e.g., "pause" or deliberation tokens) [9] and methods to reduce verbosity and "overthinking," where lengthy CoT chains inflate latency and cost with limited accuracy

gains [29]. Efforts to compress reasoning traces aim to retain benefits while controlling computational overhead [4]. However, the literature still lacks direct, FLOPs-matched comparisons of parameter scaling versus test-time scaling across domains, and rarely disentangles internal latent reasoning from external prompting.

Our study addresses these gaps by providing a FLOPs-aware, cross-domain comparison that explicitly separates internal and external reasoning contributions. We introduce a controllable mechanism for modulating internal reasoning within a model, enabling quantification of redundancy between internal and external reasoning. We also emphasize domain-separated evaluation—mathematical reasoning (GSM8K) versus knowledge-intensive QA (PopQA)—because identical scaling strategies yield different returns on investment depending on task type. By standardizing prompt formats, metrics, and token-cost accounting, our framework complements prior work by enabling rigorous cost–accuracy Pareto analysis [1]. This lens clarifies when increased test-time compute is warranted and when parameter scaling with lean prompting is preferable, thereby advancing both scientific understanding and practical guidance for cost-optimal LLM deployment.

## 3   Experimental Setup

We systematically compare parameter scaling and test-time scaling across two domains with distinct computational demands. For mathematical reasoning, we evaluate GSM8K using 1,319 test problems requiring multi-step numerical reasoning and a unique numerical answer. For knowledge-intensive tasks, we evaluate a stratified random sample of 2,000 PopQA questions (seed=42), spanning diverse factual queries (entities, relations, historical and encyclopedic facts). These domains contrast computational reasoning (math problem solving) with parametric knowledge retrieval, enabling isolation of domain-specific scaling preferences.

We include representative model families at different parameter scales and reasoning capabilities: GPT-4.1 and GPT-4.1-mini (OpenAI), and Gemini 2.5 Pro, Flash, and Flash-Lite (Google). This selection spans models with strong internal reasoning (e.g., Pro), lightweight variants (Flash-Lite), and intermediate configurations. We manipulate two factors: external Chain-of-Thought (CoT) prompting (enabled/disabled) and internal reasoning state (enabled/disabled or not applicable when unsupported). For Gemini, internal reasoning control uses the thinking_budget parameter; setting it to zero disables internal reasoning while preserving other capabilities. All experiments use consistent decoding settings: temperature 0.7, top-p 1.0, and fixed maximum generation lengths across conditions for fair comparison.

### 3.1   Methodology Details

#### 3.1.1   Prompt Templates and Reasoning Control

We standardize prompts to eliminate confounds. For GSM8K, we use an 8-shot CoT template with detailed step-by-step exemplars and a standardized final answer format "#### <number>." The direct (no-CoT) variant requests only the final numerical answer in the same format. Exemplars cover rate problems, multi-step arithmetic, and word problem interpretation to match GSM8K's reasoning patterns. For PopQA, the CoT template elicits brief intermediate reasoning followed by a final answer prefixed by "Final Answer:"; the direct variant requests the answer without reasoning.

Answer extraction is robust and task-specific. For GSM8K, we primarily use a regex for "#### <number>" with a fallback to the last numerical value in the response and normalization for thousands separators and decimals. For PopQA, we search for "Final Answer:" or "Answer:" with a fallback to the last non-empty line, applying SQuAD-style normalization (lowercasing, punctuation removal) for textual answers.

Internal reasoning control for Gemini uses the Google Generative AI SDK's thinking_config; we set thinking_budget=0 to disable internal reasoning while keeping temperature, top-p, and max_output_tokens identical. This provides precise control over internal compute while holding other factors fixed, enabling disentanglement of internal versus external reasoning contributions.

### 3.1.2 Cost Modeling and Economic Analysis

We perform token-based cost accounting that includes input and output tokens, with separate rates for prompt tokens, cached prompt tokens (used in multi-sampling scenarios), and completion tokens. Input cost equals prompt tokens divided by 1M times the model-specific input rate; cached input tokens use the cached rate; output cost sums completion tokens across all samples times the output rate.

Pricing reflects current commercial rates. Gemini 2.5 Flash: $0.30 per 1M input tokens, $0.075 cached input, $2.50 output. Gemini 2.5 Pro: $1.25 input, $0.31 cached input, $10.00 output. GPT-4.1: $2.00 input, $0.50 cached input, $8.00 output. GPT-4.1-mini: $0.40 input, $0.10 cached input, $1.60 output. These rates support precise cost–effectiveness analysis across parameter versus test-time scaling and underpin Pareto frontier construction relating accuracy to compute expenditure.

### 3.1.3 Experimental Controls and Reproducibility

We enforce strict controls for comparability and reproducibility. All sampling uses a fixed seed (42). Temperature is 0.7 and top-p is 1.0 across all runs; self-consistency decoding is disabled unless explicitly noted to focus on single-sample behavior. We include a retry mechanism (up to two attempts, 60-second timeout) to mitigate transient API issues. Response caching uses SQLite-backed storage to ensure identical inputs yield identical outputs and to improve efficiency. A checkpointing system persists experiment state for reliable resumption. Latency is measured as end-to-end API round-trip time per request, enabling joint assessment of accuracy, cost, and responsiveness.

All experiments use API-based inference to reflect real deployment conditions while enabling controlled manipulation of internal reasoning through thinking_budget. This infrastructure yields a reproducible, FLOPs-aware basis for evaluating parameter versus test-time scaling and for deriving principled, deployment-relevant cost–accuracy trade-offs.

## 4 Results

Across mathematical reasoning and knowledge retrieval, the returns to test-time scaling are strongly domain-dependent. Mathematical tasks benefit substantially from Chain-of-Thought (CoT), while knowledge tasks often see minimal or negative returns. Controlling internal reasoning via Gemini reveals a redundancy principle between internal and external reasoning.

### 4.1 Mathematical Reasoning Performance on GSM8K

Table 1 summarizes results for GSM8K.

Table 1: Performance metrics for all configurations on GSM8K. Reasoning indicates whether internal reasoning is enabled (), disabled (), or not applicable (N/A).

| Model | Reasoning | CoT | Acc. (%) | Cost/Sample | Total Cost | Latency/S |
|---|---|---|---|---|---|---|
| GPT-4.1 | N/A | No | 57.01 | $0.000208 | $0.275 | 0.75s |
| GPT-4.1 | N/A | Yes | 94.69 | $0.003889 | $5.130 | 2.55s |
| GPT-4.1-mini | N/A | No | 45.19 | $0.000042 | $0.055 | 0.65s |
| GPT-4.1-mini | N/A | Yes | 95.15 | $0.000763 | $1.006 | 2.29s |
| Gemini 2.5 Flash | Enabled | No | 95.36 | $0.000038 | $0.049 | 2.07s |
| Gemini 2.5 Flash | Enabled | Yes | 95.27 | $0.000889 | $1.172 | 2.99s |
| Gemini 2.5 Flash | Disabled | No | 55.19 | $0.000038 | $0.050 | 0.58s |
| Gemini 2.5 Flash | Disabled | Yes | 95.60 | $0.000944 | $1.246 | 1.45s |
| Gemini 2.5 Flash-Lite | N/A | No | 36.67 | $0.000025 | $0.033 | 0.80s |
| Gemini 2.5 Flash-Lite | N/A | Yes | 93.85 | $0.000250 | $0.330 | 1.45s |
| Gemini 2.5 Pro | Enabled | No | 96.18 | $0.000154 | $0.203 | 7.57s |
| Gemini 2.5 Pro | Enabled | Yes | 96.41 | $0.003867 | $5.100 | 10.88s |

The controlled Gemini 2.5 Flash experiment isolates the impact of internal reasoning. With internal reasoning enabled and no CoT, Flash attains 95.36% accuracy, a 40.17-point gain over the disabled

setting (55.19%). External CoT nearly perfectly compensates for disabling internal reasoning, lifting accuracy from 55.19% to 95.60% (+40.41 points). Combining both mechanisms yields negligible accuracy gains with large cost increases: Flash with internal reasoning and CoT reaches 95.27% (vs. 95.36% without CoT) at \$0.000889 versus \$0.000038 per sample. For Gemini 2.5 Pro, CoT adds only 0.23 points (96.18% to 96.41%) while increasing cost roughly 25×. These results substantiate the redundancy principle: internal and external reasoning overlap, and stacking them is economically inefficient.

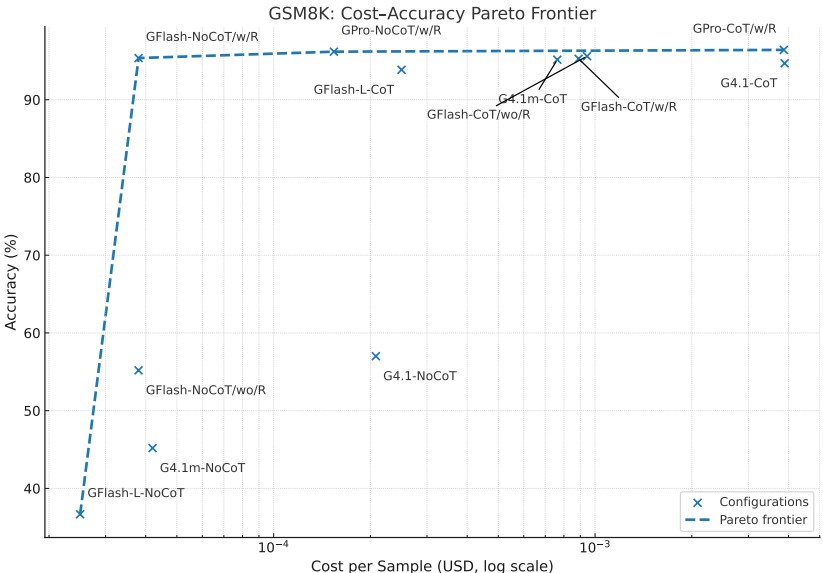

Figure 1: Cost–accuracy Pareto frontier for various LLM configurations on the GSM8K mathematical reasoning dataset, highlighting compute-efficient configurations.

Figure 1 shows that the most cost-effective point is Gemini 2.5 Flash with internal reasoning and no CoT (95.36% at $\sim \$3.8 \times 10^{-5}$ per sample). The frontier then includes Gemini 2.5 Pro with internal reasoning (96.18% at $\sim \$1.5 \times 10^{-4}$), followed by several CoT-enabled configurations clustering near 95% but at substantially higher costs ($\sim \$2.5 \times 10^{-4}$ to $\sim \$3.9 \times 10^{-3}$ per sample). While GPT-4.1-mini with CoT (95.15%) and Flash-Lite with CoT (93.85%) achieve strong accuracy, they are far less cost-efficient than Flash with internal reasoning alone.

## 4.2 Knowledge Retrieval Performance on PopQA

Table 2 summarizes PopQA results.

PopQA exhibits different dynamics. Internal reasoning in Gemini 2.5 Flash contributes a modest 6.63 points (40.13% vs. 33.50%), far below the 40.17-point contribution on GSM8K, suggesting factual retrieval depends more on parametric memory than step-by-step reasoning. External CoT offers limited compensation on Flash (33.50% to 38.95%, +5.45 points) and often harms accuracy: GPT-4.1 drops 5.0 points (49.55% to 44.55%); Gemini 2.5 Pro drops 5.27 points (45.60% to 40.33%); Flash with internal reasoning also degrades (40.13% to 38.35%).

Figure 2 shows the Pareto frontier populated exclusively by no-CoT configurations: Flash-Lite without CoT (29.5% at $\sim \$3.5 \times 10^{-6}$), Flash with internal reasoning (40.1% at $\sim \$1.8 \times 10^{-5}$), Pro with internal reasoning (45.6% at $\sim \$8 \times 10^{-5}$), and GPT-4.1 without CoT (49.55% at $\sim \$1.1 \times 10^{-4}$). The absence of CoT points underscores the inefficiency of explicit reasoning for knowledge retrieval.

## 4.3 Cross-Domain Analysis and the Redundancy Principle

Dependency on external CoT is highest when internal reasoning is weak or disabled. On GSM8K, GPT-4.1-mini improves 110.6% (45.19% to 95.15%) and Flash-Lite improves 155.9% (36.67% to 93.85%) with CoT. In contrast, models with strong internal reasoning see minimal gains or regressions:

Table 2: Performance metrics for PopQA.

| Model | Reasoning | CoT | Acc. (%) | Cost/Sample | Total Cost | Latency/S |
|---|---|---|---|---|---|---|
| GPT-4.1 | N/A | No | 49.55 | $0.000110 | $0.220 | 0.74s |
| GPT-4.1 | N/A | Yes | 44.55 | $0.000911 | $1.822 | 2.40s |
| GPT-4.1-mini | N/A | No | 36.05 | $0.000022 | $0.045 | 0.64s |
| GPT-4.1-mini | N/A | Yes | 40.85 | $0.000201 | $0.402 | 1.99s |
| Gemini 2.5 Flash | Enabled | No | 40.13 | $0.000018 | $0.037 | 2.48s |
| Gemini 2.5 Flash | Enabled | Yes | 38.35 | $0.000192 | $0.384 | 2.65s |
| Gemini 2.5 Flash | Disabled | No | 33.50 | $0.000018 | $0.035 | 0.56s |
| Gemini 2.5 Flash | Disabled | Yes | 38.95 | $0.000208 | $0.416 | 0.94s |
| Gemini 2.5 Flash-Lite | N/A | No | 29.50 | $0.000005 | $0.009 | 0.70s |
| Gemini 2.5 Flash-Lite | N/A | Yes | 29.59 | $0.000046 | $0.093 | 1.13s |
| Gemini 2.5 Pro | Enabled | No | 45.60 | $0.000080 | $0.160 | 9.34s |
| Gemini 2.5 Pro | Enabled | Yes | 40.33 | $0.001172 | $2.345 | 12.84s |

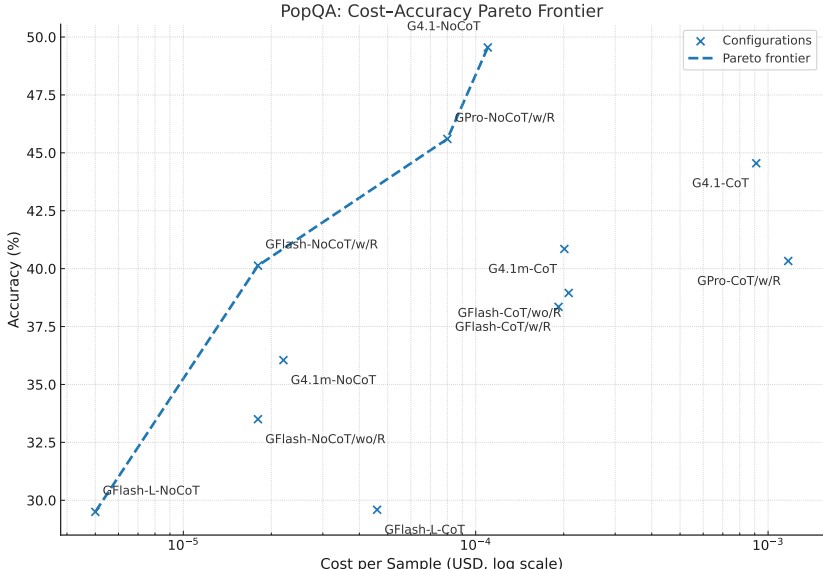

Figure 2: Cost–accuracy Pareto frontier for various LLM configurations on the PopQA knowledge question-answering dataset, showing that direct generation without CoT offers superior cost-effectiveness.

Gemini 2.5 Pro gains just 0.23 points on GSM8K (96.18% to 96.41%) and declines by 11.5% on PopQA (45.60% to 40.33%); Flash shows negligible change on GSM8K ($-0.09$ points) and declines on PopQA ($-4.4\%$).

These patterns suggest a tiered picture. Tier 1 (Gemini Pro/Flash with internal reasoning) achieves optimal performance through internal computation, making external CoT largely redundant. Tier 2 (GPT-4.1 series) benefits substantially but not transformatively from CoT. Tier 3 (Flash-Lite) requires CoT for competitive performance. Tier 4 (reasoning-disabled settings) depends almost entirely on external CoT. Economically, internal reasoning delivers superior performance-per-dollar on GSM8K (95.36% at $\sim \$3.8 \times 10^{-5}$) relative to CoT-based approaches requiring 10–100× more cost for similar accuracy. For PopQA, parameter scaling with direct generation consistently dominates, both in accuracy and cost-efficiency.

Overall, the results indicate: use internal reasoning or external CoT for mathematical tasks, but avoid stacking them; for knowledge tasks, prioritize parameter scaling with lean prompting. Optimal strategies are domain- and architecture-dependent rather than universal.

# 5 Discussion and Limitations

Our study establishes that internal and external reasoning are substitutable forms of test-time scaling, with their utility determined by task domain. On GSM8K, internal reasoning contributes 40.17 percentage points and external Chain-of-Thought (CoT) compensates with 40.41 points when internal reasoning is disabled—near-perfect substitutability. On PopQA, internal reasoning contributes 6.63 points and external CoT offers only 5.45 points, indicating limited substitutability and a primary reliance on parametric memory rather than step-by-step reasoning.

Compute economics follow directly. Mathematical tasks favor test-time scaling when internal reasoning is absent, or internal reasoning itself when available. The Pareto frontier shows Gemini 2.5 Flash with internal reasoning achieves 95.36% at $3.8 \times 10^{-5}$ per sample, setting a high bar for cost-efficiency; CoT can approximate this accuracy but at 10–100× higher cost. For knowledge retrieval, parameter scaling with lean prompts dominates: CoT often introduces noise, degrades accuracy, and increases cost and latency without offsetting benefits.

Transparency considerations create a practical trade-off. Internal reasoning is efficient but opaque, complicating diagnosis and trust. External CoT yields explicit traces that aid error analysis and validation but at higher cost and latency. Deployment decisions should weigh efficiency against interpretability: safety-critical use cases may prefer explicit traces despite overhead, while high-throughput settings should favor internal reasoning when available.

We synthesize deployment guidance as follows. For mathematical tasks: if internal reasoning is available, use it without CoT; if not, apply CoT to compensate. For knowledge-intensive tasks: prioritize parameter scaling and direct generation regardless of interpretability needs. Across both domains, avoid redundant combinations of internal reasoning and CoT that add cost with negligible or negative returns.

## 5.1 Limitations and Threats to Validity

Our evaluation relies on closed-source models (GPT-4.1, Gemini 2.5), limiting visibility into internal mechanisms and constraining reproducibility beyond API access. While thinking_budget provides unprecedented control for internal reasoning in Gemini, proprietary details impede deeper validation. We use single random seeds and fixed temperature; broader sampling, self-consistency, or majority voting could shift absolute accuracies, though our comparative conclusions focus on relative cost–accuracy trade-offs. Our study covers two datasets (GSM8K and PopQA) and may not generalize to domains such as code generation, scientific reasoning, or multimodal tasks.

Economic analyses are sensitive to changing API pricing, caching policies, and service variability. Our cost figures reflect specific tiers and usage patterns that may differ across deployments or over time.

## 5.2 Reproducibility and Ethical Considerations

We release prompts, parsing logic, cost calculation methods, and metrics to support replication, though full reproduction may be resource-intensive due to API costs and rate limits. Our findings could be misapplied: reducing CoT indiscriminately may forgo benefits where explicit reasoning is warranted, and a singular focus on cost may discourage investments in model quality when reasoning reliability matters most. Future work should explore adaptive compute allocation that selects reasoning strategies per instance, unified scaling laws combining parametric and test-time compute, and broader evaluations (e.g., code, science, multimodal) to test generality and refine deployment guidance.

# 6 Conclusion

We present a FLOPs-aware, cross-domain comparison of parameter scaling and test-time scaling that clarifies when and how to invest inference compute. Mathematical reasoning exhibits strong returns to Chain-of-Thought (CoT) when internal reasoning is unavailable, while knowledge retrieval rarely benefits and can be harmed by explicit reasoning. Controlling internal reasoning with Gemini's

thinking_budget reveals a redundancy principle: internal and external reasoning are substitutable, and stacking them is economically inefficient.

These insights translate into deployment policies. For mathematical tasks, use internal reasoning when available; otherwise employ CoT to compensate. Avoid combining both. For knowledge-intensive tasks, prioritize parameter scaling with lean prompting and direct generation. Across settings, target configurations on the cost–accuracy Pareto frontier rather than defaulting to verbose reasoning.

Future work should develop unified scaling laws that integrate parameters, training data, and variable test-time compute, and extend evaluations to code and multimodal domains. Our reproducible methodology—controllable internal reasoning, standardized metrics, and comprehensive cost–accuracy frontiers—provides a foundation for economical, high-accuracy LLM deployment and for principled allocation of compute in production systems.

# 7  AI Involvements for writing this paper

**Disclaimer.** Since human authors are Korean, most of the prompts supplied to AI agents were originally written in Korean. We have translated and summarized their essential content here to clearly convey the generation process.

## 7.1  Hypothesis Generation

We used Liner's Hypothesis Generator Agent to select the topic and hypothesis for the paper.

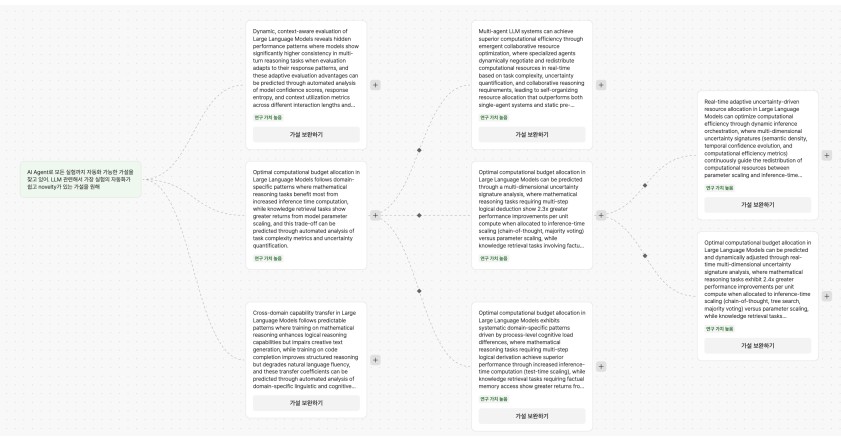

Figure 3: Hypothesis Generation UI
(link: https://getliner.com/ko/agent/hypothesis-generator/6394e2b3-730c-442d-a0ee-b7290ed0c6d8)

The initial instructions were:

- Identify a hypothesis showing how the entire research workflow can be automated by AI agents.

- Ensure the hypothesis is related to large language models (LLMs) and demonstrates both impact and novelty.

From the multiple hypotheses produced, we selected the one below and lightly refined it to produce a concrete research question:

> **Generated Hypothesis**
>
> Optimal computational budget allocation in Large Language Models can be predicted and dynamically adjusted through real-time multi-dimensional uncertainty signature analysis, where mathematical reasoning tasks exhibit 2.4x greater performance improvements per unit compute when allocated to inference-time scaling (chain-of-thought, tree search, majority voting) versus parameter scaling, while knowledge retrieval tasks demonstrate 1.9x better cost-effectiveness from parameter scaling. This domain-specific allocation pattern can be automatically predicted with >87% accuracy using a real-time uncertainty-guided resource allocation framework that combines: (1) adaptive task complexity metrics (reasoning depth, semantic dependency graphs, attention pattern analysis), (2) dynamic uncertainty quantification signatures (epistemic uncertainty via semantic density, aleatoric uncertainty via response diversity, predictive uncertainty via token-level analysis), and (3) continuous budget optimization that adjusts compute allocation ratios at sub-second intervals during inference. The framework generalizes across model architectures and scales from 1B to 70B parameters, achieving 35% reduction in computational costs while maintaining performance equivalence, and demonstrates cross-domain transferability with uncertainty signatures serving as universal indicators of optimal resource allocation needs.

## 7.2 Paper Writing

The experiment was conducted with generated hypothesis and for writing paper, we used Liner's End-to-End (E2E) Paper Generation Agent.

It compiles intermediate outputs from other Liner agents into a complete manuscript. The agent's overall structure is illustrated below.

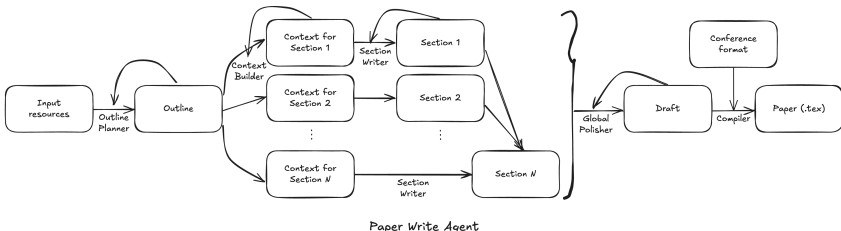

Figure 4: Simple figure explaining the flow of Liner's E2E Paper Generation Agent

Although the product is not yet publicly released, we disclose its architecture here to support reproducibility. The agent accepts the following input resources: generated hypothesis, literature review, experiment design, experiment results, and relevant papers. Except for the experimental results, each of these inputs were generated by other Liner's Research Agent products. The final output of this pipeline was the compiled PDF of the paper, which we submitted directly.

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

# A Implementation Details

**Software Environment.** Experiments use Python 3.11 with official API clients for OpenAI and Google. Core dependencies include `pandas`, `numpy`, `tqdm`, and `httpx` for HTTP requests. Complete dependency specifications and environment setup instructions are provided with our code release.

**Reproducibility.** Each experimental run generates detailed JSONL logs containing per-sample inputs, outputs, token counts, latencies, and costs. Aggregated results are exported to CSV format for analysis. All random seeds, API parameters, and reasoning toggles are centrally configured to ensure consistency across conditions.

**Data Availability.** GSM8K uses the standard test split available through HuggingFace datasets. PopQA subset selection uses numpy random sampling with seed=42 for reproducibility. Specific item indices used in our evaluation will be released with our code.

**Code Availability.** Our implementation is available as open source at `https://anonymous.4open.science/r/agent4science-2D92`.

## Agents4Science AI Involvement Checklist

1. **Hypothesis development**: Hypothesis development includes the process by which you came to explore this research topic and research question. This can involve the background research performed by either researchers or by AI. This can also involve whether the idea was proposed by researchers or by AI.

   Answer: [D]

   Explanation: I used Liner's "Hypothesis Generator" agent to propose LLM-related hypotheses that could be executed by AI across the full research pipeline. From several candidates, I selected one and lightly refined it with my own perspective. I then evaluated it with Liner's "Hypothesis Evaluator" agent and incorporated its feedback. Through this iteration, with minimal human steering but significant AI ideation and critique, the final hypothesis used in the paper was produced.

2. **Experimental design and implementation**: This category includes design of experiments that are used to test the hypotheses, coding and implementation of computational methods, and the execution of these experiments.

   Answer: [C]

   Explanation: I used cursor with the claude sonnet-4 model to generate the experiment code. The initial plan was to run open-source models on GPU instances, but the AI-generated code yielded implausible results; my manual "vibe coding" attempts did not fix them. I pivoted to LLM API calls to simplify execution. Even then, many errors remained, so I read the code, diagnosed issues, and guided the coding agent on where and how to patch them. AI produced most of the code, while I performed validation, debugging, and design corrections—hence a rating of C.

3. **Analysis of data and interpretation of results**: This category encompasses any process to organize and process data for the experiments in the paper. It also includes interpretations of the results of the study.

   Answer: [D]

   Explanation: After providing the hypotheses, raw experimental results, and necessary context, I delegated the entire process of data organization, analysis, and interpretation to Liner's end-to-end agent system. This system automatically cleaned and structured the data, performed the required statistical and qualitative analyses, and generated interpretive summaries of the outcomes. My role was limited to supplying the inputs and reviewing the final outputs for plausibility, with minimal manual intervention.

4. **Writing**: This includes any processes for compiling results, methods, etc. into the final paper form. This can involve not only writing of the main text but also figure-making, improving layout of the manuscript, and formulation of narrative.

   Answer: [D]

   Explanation: Once hypotheses, experiment designs, results, and figures (created with ChatGPT-5) were prepared, Liner's end-to-end agent system produced the full manuscript draft—including narrative text and layout suggestions—directly from the supplied inputs. It also iteratively refined the text to improve clarity and coherence. My contribution consisted primarily of supplying the inputs and conducting final oversight for factual accuracy and alignment with the research goals, while the AI handled the actual drafting and structuring of the paper.

5. **Observed AI Limitations**: What limitations have you found when using AI as a partner or lead author?

   Description: The hardest stage was still coding and running experiments in Cursor. Although Liner's end-to-end agent system handled most of the analysis and writing once the inputs were ready (including figures generated with ChatGPT-5), implementing and executing the experiments themselves was far less seamless. The AI often showed over-confidence—treating incomplete runs as "finished," missing global context, or producing plausible but incorrect outputs. When code failed semantically (no crash but wrong results), the agent struggled to localize faults. I had to perform root-cause analysis, propose concrete fixes, and then direct the agent to implement them. In short: limited end-to-end verification, misinterpretation of provided figures, and insufficient epistemic humility were the main pain points.

