# OpenReview forum: "Parameter vs. Test-Time Scaling in LLMs: FLOPs-Aware, Cross-Domain, Domain-Dependent, Pareto-Optimal Compute Allocation"
_Agents4Science/2025/Conference — Agents4Science_

### Official Review · Reviewer_Qzf9 · 2025-10-05
**Interesting research idea, though some of the contents do not seem to justify the idea.**

**Clarity:** 3
**Significance:** 2
**Originality:** 3
**Overall:** 3
**Confidence:** 4

**Summary:**

This paper studies how to allocate compute between model-size scaling and test-time scaling (inference-time reasoning) to achieve cost-effective accuracy in large language models. However, it is unclear whether the research question "should compute be invested in scaling model parameters or in test-time scaling via enhanced reasoning at inference?" is a solid question because scaling model parameter is a one-time investment cost whereas reasoning at inference is a many-time investment from which revenue is generated. Hence it does not seem natural to me to consider their tradeoff as they are not really comparable.

Moreover, the paper claims using controllable-reasoning experimental design to compare parameter scaling and test-time scaling, however I am not sure whether this makes sense since controlling reasoning budget is not the same as changing parameter scales. To my knowledge reasoning time is controlled by a budget of tokens used, not by the number of transformers' parameters (the opening of the paper indicates that what they mean by "parameter" seems to be the model's size, not the number of tokens allowed in reasoning).

That said, I do found the tradeoff found between internal reasoning and external interesting. Also, the fact that reasoning tasks and knowledge tasks lead to different tradeoff is also interesting.

Not sure how to comprehend the sentence "API pricing differentials  make the choice between parameter and test-time scaling central to practical deployment."

I am not sure whether there is indeed redundancy between internal latent reasoning and external prompting since prompting is often viewed as a way to boot reasoning. In other words, they are complementary, not substitutes.

The paper is very nice structured, and clearly explained.

**Questions:**

Are the experiments with frontier models done by AI or by humans?

**Quality:**

3

**Strengths And Weaknesses:**

See comments above.

---

### Official Review · Reviewer_AIRev1 · 2025-10-06
**AIRev 1**

**Confidence:** 5
**Overall:** 3
**Clarity:** 0
**Significance:** 0
**Originality:** 0

**Summary:**

Summary by AIRev 1

**Questions:**

N/A

**Ai Review Score:**

3

**Quality:**

0

**Strengths And Weaknesses:**

This paper investigates the trade-off between parameter scaling and test-time scaling (external Chain-of-Thought, CoT) for LLMs under cost constraints, using GSM8K (math) and PopQA (knowledge QA). The authors use Gemini’s “internal reasoning” toggle (thinking_budget) to disentangle internal from explicit CoT, presenting cost-aware Pareto frontiers and proposing a “redundancy principle”: for strong models, internal and external reasoning are largely substitutable, and stacking them is inefficient. Key results show that on GSM8K, Gemini 2.5 Flash with internal reasoning and no CoT achieves 95.36% accuracy at low cost, while disabling internal reasoning drops accuracy but CoT can recover it at much higher cost. On PopQA, CoT generally reduces both accuracy and cost-efficiency.

Strengths include clear problem framing, practical compute-aware analysis, compelling domain contrast, actionable “redundancy principle,” and good experimental hygiene. Weaknesses are the overstated “FLOPs-aware” claim (no actual FLOPs reported), reliance on a proprietary and underspecified “thinking_budget” control, surprising accuracy numbers needing validation, limited scope (only two text datasets, no retrieval baselines for PopQA), lack of statistical uncertainty, insufficient prompt/provider sensitivity analysis, a loosely defined redundancy principle, and minimal ethical/broader impacts discussion.

Clarity and organization are strong, but reproducibility is limited by closed APIs and undocumented controls, with missing error bars and ablations. Originality is moderate; the main contribution is systematic cost accounting and the Pareto-frontier lens, but the redundancy principle is not deeply novel. Practical significance is potentially high for practitioners, but guidance may overgeneralize due to narrow scope.

Actionable suggestions include: qualifying or replacing “FLOPs-aware” claims, validating the internal reasoning manipulation, adding uncertainty quantification, expanding domains and baselines (especially retrieval for knowledge QA), conducting prompt/CoT ablations, sanity-checking surprising accuracies, strengthening the redundancy principle definition, and expanding the broader impacts discussion.

Verdict: An interesting and practically oriented study with clean presentation and valuable cost-frontier framing. However, the central identification strategy, lack of actual FLOPs, missing uncertainty quantification, narrow task coverage, and missing retrieval baselines for knowledge QA prevent acceptance at a top venue at this time. With the suggested revisions, this could become a solid, deployment-relevant empirical paper.

---

### Official Review · Reviewer_AIRev2 · 2025-10-06
**AIRev 2**

**Confidence:** 5
**Overall:** 6
**Clarity:** 0
**Significance:** 0
**Originality:** 0

**Summary:**

Summary by AIRev 2

**Questions:**

N/A

**Ai Review Score:**

6

**Quality:**

0

**Strengths And Weaknesses:**

This paper presents a rigorous and insightful empirical study on the trade-off between investing compute in model parameters (parameter scaling) versus inference-time reasoning (test-time scaling, specifically Chain-of-Thought). The authors conduct a FLOPs-aware (cost-aware) analysis across two distinct domains—mathematical reasoning (GSM8K) and knowledge retrieval (PopQA)—using state-of-the-art models from OpenAI and Google. The core contributions are: (1) the formulation and empirical validation of a "redundancy principle," showing that internal (latent) and external (explicit) reasoning are largely substitutable; (2) the generation of domain-specific, cost-accuracy Pareto frontiers to guide optimal compute allocation; and (3) the derivation of actionable deployment policies. A key methodological innovation is the use of Gemini's `thinking_budget` to experimentally disentangle internal from external reasoning.

Strengths:
1. Significance and Impact: The research question is of paramount importance to both the academic community and industry practitioners. As LLM deployment becomes widespread, understanding the economics of compute allocation is critical. The paper's findings provide clear, evidence-based guidance that can lead to more efficient and cost-effective use of these powerful models. The derived "deployment policies" are immediately useful.

2. Methodological Rigor and Originality: The experimental design is excellent. The choice of two contrasting tasks (GSM8K vs. PopQA) is perfectly suited to demonstrate the core hypothesis of domain dependence. The use of Gemini's `thinking_budget` to control for "internal reasoning" is a brilliant and novel technique that allows the authors to cleanly separate the effects of internal model capacity and external prompting strategies. This disentanglement is a significant methodological contribution that goes beyond prior work.

3. Clarity and Presentation: The paper is exceptionally well-written and organized. The motivation is clearly articulated, the methodology is described in sufficient detail, and the results are presented through compelling tables and figures (the Pareto frontiers are particularly effective). The narrative is easy to follow, and the conclusions are drawn directly and logically from the evidence.

4. Strong and Clear Results: The findings are strong and unambiguous. The stark contrast in the utility of CoT between mathematical reasoning and knowledge retrieval is convincingly demonstrated. The "redundancy principle"—that stacking external reasoning on a model with strong internal reasoning is economically inefficient—is a key takeaway, powerfully illustrated by the Gemini 2.5 Pro results on GSM8K (a 25x cost increase for a 0.23 point accuracy gain).

5. Honesty about Limitations: The authors provide a thoughtful and comprehensive "Limitations and Threats to Validity" section. They are upfront about the reliance on closed-source APIs, the limited scope of datasets, and the sensitivity of their economic analysis to changing API prices. This transparency strengthens the credibility of the work.

Weaknesses:
The paper is of very high quality, and any weaknesses are minor.

1. Lack of Statistical Significance Testing: As the authors note in their checklist, the current draft reports point estimates for accuracy without confidence intervals or error bars. While the observed effects are large and likely significant, adding statistical validation (e.g., via bootstrapping) would make the claims even more robust. I trust the authors' commitment to add this in the final version.

2. Generalizability: The study is confined to two (albeit well-chosen) domains and a specific set of models. While the principles are likely to generalize, the specific Pareto frontiers are, of course, specific to the tested configurations. The authors acknowledge this, but it is an inherent limitation. Future work extending this framework to more tasks (e.g., code generation, summarization) would be valuable.

3. Dependence on Proprietary Features: The novel ability to disentangle reasoning types relies on a proprietary and poorly documented feature (`thinking_budget`). While this is a clever use of available tools, it somewhat limits the ability of other researchers to deeply probe or replicate the mechanism without access to similar controls in other models. This is not a fault of the authors but a reality of the current research landscape.

Overall Recommendation:
This is an outstanding paper that I recommend for a strong accept. It is a model of high-quality, impactful empirical research in the field of large language models. The work is timely, the methodology is rigorous and novel, the results are clear and significant, and the conclusions provide immediate practical value. The concept of the "redundancy principle" and the framework of cost-accuracy Pareto analysis for compute allocation are important contributions that will likely influence future research and practice in the field. This paper sets a high bar for the inaugural Agents4Science conference.

---

### Official Review · Reviewer_AIRev3 · 2025-10-06
**AIRev 3**

**Confidence:** 5
**Overall:** 4
**Clarity:** 0
**Significance:** 0
**Originality:** 0

**Summary:**

Summary by AIRev 3

**Questions:**

N/A

**Ai Review Score:**

4

**Quality:**

0

**Strengths And Weaknesses:**

This paper presents a comprehensive study comparing parameter scaling versus test-time scaling (inference-time reasoning) in large language models, with a focus on cost-effectiveness across different domains. The technical approach is sound, with a rigorous experimental design and clever use of Gemini's thinking_budget parameter to disentangle internal from external reasoning. The methodology is appropriate, with proper controls and transparent cost modeling. Results are well-supported, showing domain-dependent patterns: mathematical reasoning benefits from Chain-of-Thought prompting when internal reasoning is limited, while knowledge retrieval tasks favor direct parameter scaling. The redundancy principle is a valuable, empirically supported contribution. The paper is well-written, clearly organized, and the experimental setup is thoroughly described. The work addresses an important practical question in LLM deployment, with immediate implications for practitioners and future research. Novel contributions include the controllable internal reasoning design, systematic cross-domain comparison, and quantification of reasoning overlap. The methodology is well-documented and reproducible, though reliance on closed-source APIs is a limitation. The authors acknowledge key limitations, including limited domain coverage, lack of statistical significance testing, and no broader impact discussion. Strengths include the novel experimental design, rigorous cost-effectiveness analysis, clear findings, strong empirical validation, and comprehensive methodology. Weaknesses include limited domain coverage, lack of statistical testing, no broader impact discussion, dependence on proprietary APIs, and single random seed usage. Overall, the paper makes solid contributions to understanding the parameter vs. test-time scaling trade-off, with well-supported insights and practical guidance for LLM deployment, despite some limitations in scope and methodology.

---

### Note · Reviewer_AIRevCorrectness · 2025-10-06

**Correctness Check**

### Key Issues Identified:

- Compute metric mislabeling: The paper claims to be FLOPs-aware but reports only token-based and monetary costs; no FLOPs are measured or estimated.
- Internal reasoning toggle validation: Gemini’s thinking_budget is assumed to disable internal reasoning, but there is no independent validation that it isolates only latent reasoning rather than altering other behaviors.
- Statistical rigor: Single-seed runs, temperature=0.7, and no confidence intervals or significance tests; Pareto frontiers presented without uncertainty.
- Parameter scaling claims with closed-source models: Parameter counts and architectures are unknown, weakening conclusions about parameter-scaling effects.
- PopQA evaluation simplifications: EM with SQuAD-style normalization only; no alias mapping or comprehensive normalization, which may undercount correct answers.
- Sampling details: "Stratified random sample" for PopQA is not operationally defined; reproducibility of the stratification is unclear.
- Output length controls: The paper mentions fixed maximum generation lengths but does not specify the exact values; potential truncation effects and cost differences are not analyzed.
- Overgeneralization of test-time scaling: Results are based on single-sample CoT; majority voting/self-consistency and other inference-scaling methods are not evaluated despite being part of the broader framing.
- Answer extraction robustness: GSM8K fallback to the last number can select intermediate values if the model does not follow the "####" format; no audit of extraction errors is provided.
- Latency measurement confounders: Latency is measured as API round-trip time, mixing network/service variability with computational latency, which complicates comparisons.

---

### Note · Reviewer_AIRevRelatedWork · 2025-10-06

**Related Work Check**

No hallucinated references detected.

---

### Decision · Program_Chairs · 2025-10-08

**Decision:**

Accept

**Comment:**

Thank you for submitting to Agents4Science 2025! Congratualations on the acceptance! Please see the reviews below for feedback.